# The ECM and tissue architecture are major determinants of early invasion mediated by E-cadherin dysfunction

Soraia Melo [1,2], Pilar Guerrero [3], Maurício Moreira Soares [4], José Rafael Bordin [5], Fátima Carneiro[1,2,6], Patrícia Carneiro[1,2], Maria Beatriz Dias [7], João Carvalho [8], Joana Figueiredo [1,2,6,9 ✉], Raquel Seruca[1,2,6,9,10] & Rui D. M. Travasso [8,9 ✉]

Germline mutations of E-cadherin cause Hereditary Diffuse Gastric Cancer (HDGC), a highly invasive cancer syndrome characterised by the occurrence of diffuse-type gastric carcinoma and lobular breast cancer. In this disease, E-cadherin-defective cells are detected invading the adjacent stroma since very early stages. Although E-cadherin loss is well established as a triggering event, other determinants of the invasive process persist largely unknown. Herein, we develop an experimental strategy that comprises in vitro extrusion assays using E-cadherin mutants associated to HDGC, as well as mathematical models epitomising epithelial dynamics and its interaction with the extracellular matrix (ECM). In vitro, we verify that E-cadherin dysfunctional cells detach from the epithelial monolayer and extrude basally into the ECM. Through phase-field modelling we demonstrate that, aside from loss of cell-cell adhesion, increased ECM attachment further raises basal extrusion efficiency. Importantly, by combining phase-field and vertex model simulations, we show that the cylindrical structure of gastric glands strongly promotes the cell's invasive ability. Moreover, we validate our findings using a dissipative particle dynamics simulation of epithelial extrusion. Overall, we provide the first evidence that cancer cell invasion is the outcome of defective cell-cell linkages, abnormal interplay with the ECM, and a favourable 3D tissue structure.

[1] i3S – Instituto de Investigação e Inovação em Saúde, Universidade do Porto, Porto, Portugal. [2] Ipatimup – Institute of Molecular Pathology and Immunology of the University of Porto, University of Porto, Porto, Portugal. [3] Departamento de Matemáticas and Grupo Interdisciplinar de Sistemas Complejos (GISC), Universidad Carlos III de Madrid, Leganés, Spain. [4] Oslo Center for Biostatistics and Epidemiology, Faculty of Medicine, University of Oslo, Oslo, Norway. [5] Department of Physics, Institute of Physics and Mathematics, Federal University of Pelotas, Capão do Leão, Rio Grande do Sul, Brazil. [6] Department of Pathology, Faculty of Medicine, University of Porto, Porto, Portugal. [7] CISUC, Department of Informatics Engineering, University of Coimbra, Coimbra, Portugal. [8] CFisUC, Department of Physics, University of Coimbra, Coimbra, Portugal. [9] These authors contributed equally: Joana Figueiredo, Raquel Seruca, Rui D. M. Travasso. [10] Deceased: Raquel Seruca. ✉email: jfigueiredo@i3s.up.pt; ruit@uc.pt

ntercellular adhesion is crucial for developmental processes, as well as for tissue integrity and homoeostasis[1,2]. The control of cell-cell adhesion is orchestrated by calcium-dependent molecules known as Cadherins[3,4]. In epithelia, E-cadherin concentrates at adherens junctions, dynamically interacting with the actomyosin cytoskeleton through catenin complexes, and playing an important role in tissue structure, differentiation and function[3,5]. Downregulation of E-cadherin at the basolateral surface of cells is implicated on a broad range of epithelial tumours, associated with infiltrative abilities and poor disease prognosis[6,7].

The hereditary form of diffuse gastric cancer, the so called Hereditary Diffuse Gastric Cancer (HDGC, OMIM #137215), is a paradigm disease model depicting the importance of E-cadherin dysfunction in cancer cell invasion[8,9]. In this syndrome, germline mutations of the E-cadherin gene (CDH1) are causative events and, at the time of clinical diagnosis, mutation carriers frequently display advanced disease stages with diffuse invasion of the gastric wall and cancer spreading into the peritoneal cavity and adjacent organs[9,10]. In light of this silent and remarkable invasive phenotype, patients lack efficient clinical surveillance and are advised to undergo prophylactic total gastrectomy, which remains the only risk-reducing approach[11,12]. Macroscopic analyses of the stomachs from affected individuals nearly always appear normal to the naked eye, with absence of mass lesion[10,13]. However, upon close and extensive histopathological inspection of the organ, intramucosal carcinoma or isolated invasive cancer cells (signet ring cell carcinoma—SRCC) can be detected in most cases[10,13]. In fact, it is well established that isolated invasion of neoplastic cells occurs since very early stages of HDGC[14]. Precursor lesions include in situ SRCC or pagetoid spread of signet ring cells below the preserved epithelium of gastric glands (Fig. 1)[13,14]. Nonetheless, the molecular mechanisms underlying this aggressive behaviour are far from being understood.

We envision that these precursor lesions are a manifestation of the cell extrusion process. Cell extrusion has been defined as a mechanism to control cell number within epithelia and prevent the accumulation of excess cells (overcrowding)[15]. Unwanted cells undergo programmed cell death that is followed by apical extrusion or delamination, and ultimately apoptosis[15].

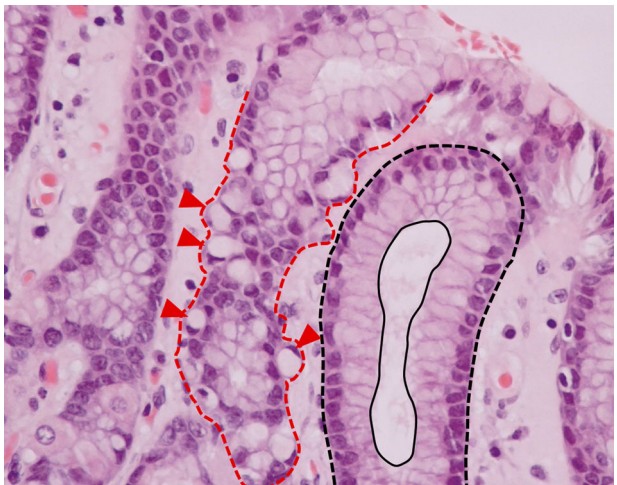

**Fig. 1 Microscopic features of HDGC.** A gastric gland presenting extrusive figures is highlighted by a red dashed line. Arrowheads indicate spreading of isolated signet-ring cells bellow the preserved epithelium of glands. In contrast, a normal gastric gland is lined by well oriented columnar cells, with nuclei localized at the base of the foveolar cells (black dashed line). The apical epithelial surface is depicted with a black continuous line.

Importantly, evidence has emerged demonstrating that cells may escape apical delamination upon oncogenic transformation[16]. Instead of being eliminated towards the lumen, transformed cells highjack this process and extrude in a basal direction into the stroma, subsequently proliferating and invading adjacent tissues[16]. During invasion, cells migrate and overpass the extracellular matrix (ECM) through morphological adaptation, activation of specific signalling cascades, and proteolytic degradation of matrix components.

In the last years, computational methods have been developed to model complex biological systems and were found to be valuable tools to identify novel regulatory mechanisms at the intracellular level[17,18], and to model the interaction between cells and their micro-environment[19–22]. In the scope of cell adhesion and motility, phase-field and vertex models, in particular, have been used with success to decode cell morphological alterations and the role played by mechanical forces during cell motion[23,24]. Indeed, it was recently shown that the density of matrix fibres modulate cancer cell shape and migration velocity in a mechanism dependent on myosin contraction[25].

In this work, we hypothesised that the interaction between E-cadherin-deficient cells and the ECM is determinant for gastric cancer development. This hypothesis is supported by data demonstrating that loss of cell polarity mediated by E-cadherin impairment changes the area and the molecules (type and quantity) involved in the cell-ECM interplay[26,27]. Moreover, physical and signalling properties of ECM can activate specific mechano-transduction programs that culminate in basal extrusion of malignant cells – the initial step in the invasive process[15,28].

To advance on the effects mediated by E-cadherin dysfunction at epithelia, we took advantage of in vitro assays using E-cadherin mutants associated to HDGC, coupled with mathematical modelling of cell-cell and cell-matrix interactions. Herein, we have engineered an epithelial monolayer cultured on top of collagen to monitor the behaviour of E-cadherin mutant cells in a wild-type context. A computational pipeline, encompassing a novel phase-field model and a vertex simulation, was then implemented to investigate cell fate upon loss of cell-cell contact and exposure to different degrees of ECM attachment. With this approach, we have also explored the contribution of tissue architecture to evasion of mutant cells from the normal epithelium. Results were further validated with a dissipative particle dynamics (DPD) simulation of cell movement and adhesion. Overall, we have demonstrated that the extrusion phenotype of HDGC relies on increased attachment capacities of E-cadherin dysfunctional cells and on the cylindrical structure of gastric glands.

## Results

**E-cadherin dysfunction grants cells increased extrusive abilities**. The process through which E-cadherin defective cells adapt and attach to the ECM in order to invade remains a major cancer research issue. To determine the ability of E-cadherin dysfunctional cells to detach from the epithelial monolayer and penetrate the ECM, we have first established a panel of cancer cell lines expressing E-cadherin missense variants associated with HDGC. To exclude possible site-dependent effects, we have chosen the A634V, R749W, and V832M E-cadherin mutants, which affect respectively the extracellular, juxtamembrane and intracellular domains of the protein. E-cadherin mutant cells labelled with a fluorescent dye were mixed with wild-type cells at highly diluted ratios and cultured on top of a collagen matrix, creating a monolayer system in which one mutant cell is surrounded by wild-type neighbours (Fig. 2a).

In contrast to a continuous epithelium formed by E-cadherin defective cells, this cellular system mimics the random

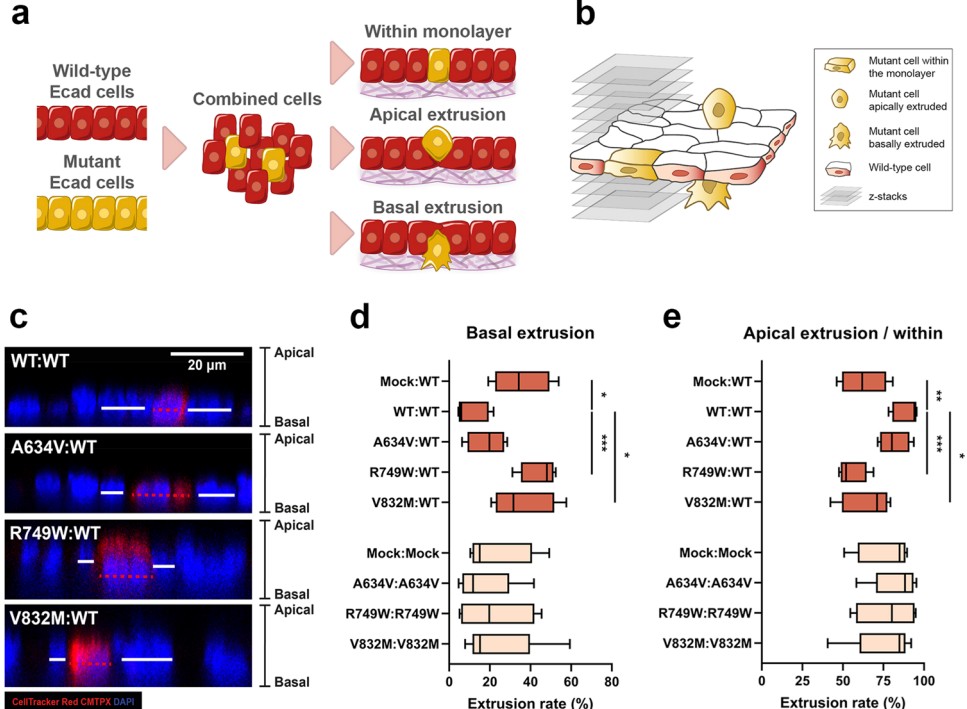

**Fig. 2 Extrusive capacity of E-cadherin mutant cells. a** Schematic representation of the methodology used to evaluate extrusive capability of E-cadherin mutants within an in vitro monolayer. **b** A *z*-stack based strategy was applied to evaluate cell nucleus position. **c** Confocal microscopy images of wild-type or mutant E-cadherin cells when surrounded by the normal context. Nuclei are counterstained with DAPI (blue). Red dashed lines in *xz* images indicate the nucleus position of labelled cells, whereas white lines represent the nucleus position of neighbouring wild-type cells. Scale bar corresponds to 20 μm. Graphs indicating **d** basal and **e** apical extrusion ratios of A634V, R749W, and V832M E-cadherin mutants in at least four independent experiments. The Mock condition corresponds to the same cell line transfected with the empty vector, and is used as a control condition devoid of E-cadherin expression and function. * corresponds to $p \leq 0.05$, ** $p \leq 0.01$, and *** $p \leq 0.001$.

appearance of E-cadherin defective cells in a normal gastric epithelium, according to the widely accepted model of isolated and diffuse spreading of tumour cells described for the early stages of HDGC[14,29].

Confocal *xz*-sections were applied to the monolayers as an estimate of the position of each cell (mutant and wild-type) relative to the normal epithelia (Fig. 2b).

As observed in Fig. 2c, the nuclei of mutant cells (shown in red) are localised basally, when compared with those of the wild-type reference that are in a more apical position. Quantification of nucleus phenotypes revealed that all E-cadherin mutants lead to an increase in basal cell extrusion. In fact, when surrounded by wild-type cells, 18.63%, 44.91%, and 35.94% of the A634V, R749W, and V832M mutant cells, respectively, basally extrude (R749W $p = 0.0006$ and V832M $p = 0.04$, Fig. 2d). Interestingly, cells expressing the juxtamembrane R749W mutant, which leads to loss of E-cadherin expression due to trafficking deregulation, are those displaying the most extrusive potential. Regarding apical delamination, the opposite effect is seen. Those conditions depicting lower basal extrusion levels are the ones with higher apical extrusion. The extracellular mutant A634V resembles the wild-type condition and exhibits increased apical extrusion levels, when compared with the juxtamembrane and intracellular mutants (Fig. 2e). This suggests that extrusion patterns depend on E-cadherin function, as well as on the E-cadherin domain affected.

**Increased attachment to the ECM promotes basal cell extrusion.** To further investigate the extrusion process mediated by loss of E-cadherin function, we implemented a three-dimensional phase-field model mimicking a flat tissue above the ECM

(Fig. 3a–c). The simulation comprised 16 cells vertically aligned and arranged in a hexagonal lattice. To epitomise HDGC tissue context and its association with E-cadherin germline mutations, we considered a single cell that is unable to adhere to its neighbouring cells. The behaviour of the mutant cell, as compared to the wild-type adhering cells, was then monitored over time. We observed that the mutant cell leaves the epithelium, penetrating the ECM layer that mimics the basement membrane (Fig. 3a–c). As the cell leaves the tissue, it extends protusions that adhere to the ECM fibres and support its extrusion (Fig. 3c). Meanwhile, the remaining cells reorganize to occupy the free space left by the mutant cell.

E-cadherin dysfunctional cells were recently described to increase traction forces and activate specific ECM receptors, resulting in an advantageous interplay with the ECM[27,28]. Therefore, to evaluate the relevance of the ECM in our system, we have next modulated mutant cell attachment to ECM fibres and measured cell extrusion distance. We verified that mutant cells with increased cell-ECM adhesion display more effective abilities to move through the ECM, which is reflected in increased travelled distances and extrusion velocity (Fig. 3d). However, when subjected to an adhesion strength equal to its neighbouring cell, the mutated cell is also able to extrude from the epithelial tissue in the basal direction, indicating that loss of E-cadherin can induce basal extrusion through either associated downstream signalling or mechanotransduction outputs. In contrast, in the absence of ECM adhesion, the mutant is apically extruded. In this situation, the neighbouring normal cells take advantage of their higher adhesion to the ECM, increasing their contact area with the fibres and slowly pushing the mutant cell in the apical direction. These observations indicate that cell attachment to the

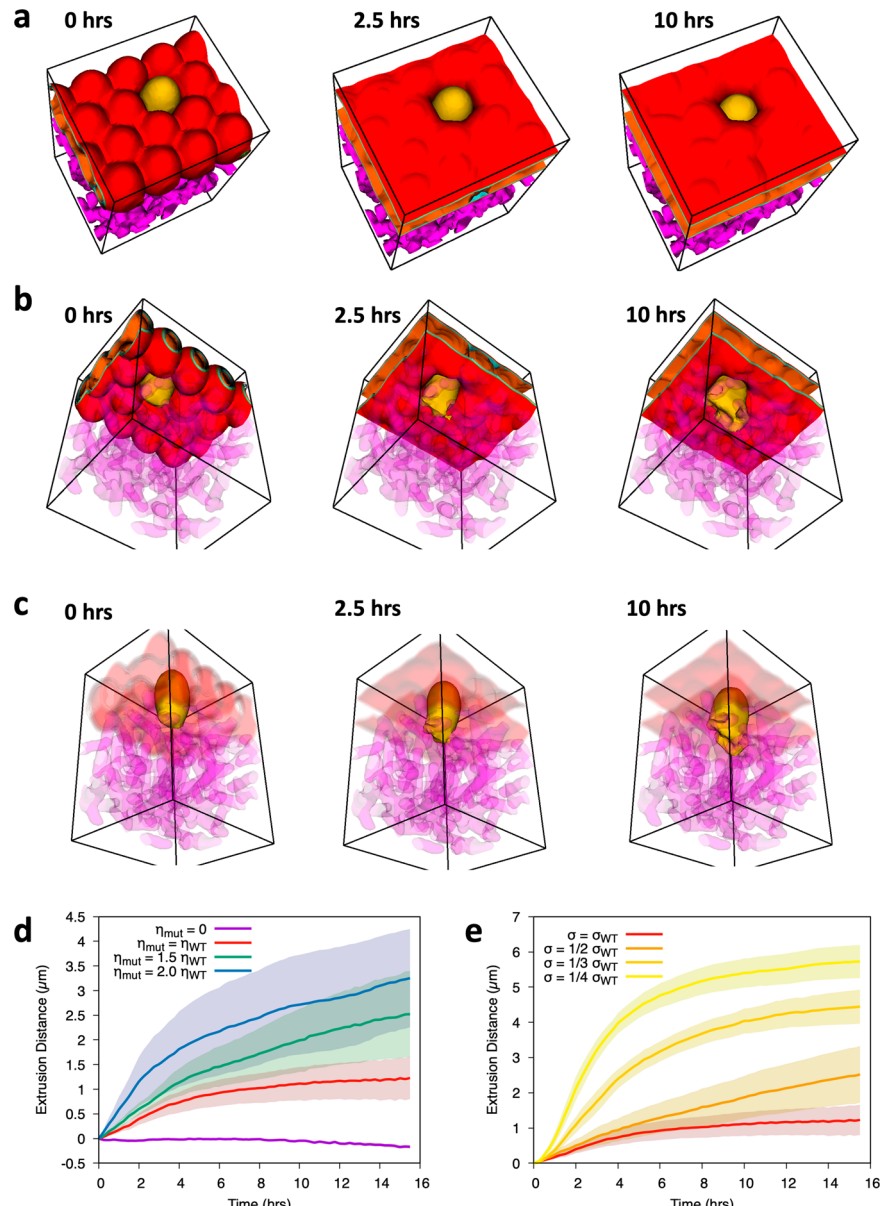

**Fig. 3 Cell-ECM adhesion promotes basal cell extrusion. a** Top view of a typical simulation run at different time points. On the left, it is shown the epithelium phenotype at $t = 0$ h; in the middle, $t = 2.5$ h; on the right, $t = 10$ h. Epithelial cells are labelled in red, the E-cadherin mutant cell is represented in yellow, and matrix fibres in magenta. **b** Basal view of the same simulation run at $t = 0$ h, 2.5 h and 10 h. **c** Representation of the same simulation run highlighting the mutant cell morphology during the extrusion process ($t = 0$ h, 2.5 h and 10 h, respectively). **d** Graph displaying average extrusion distance of the E-cadherin mutant cell along 16 h, when adhesion to ECM is 0, 1, 1.5, and 2 times that of the wild-type neighbours. **e** Graph displaying average extrusion distance of the E-cadherin mutant cell along 16 h, when extruding cell membrane tension is 1.0, 0.5, 0.33, and 0.25 times that of the cell membrane tension in wild-type neighbours. For each condition, 8 simulations were performed and standard deviations in the mean extrusion distance are represented by shaded bands.

ECM is crucial to overcome normal neighbouring pressure and initiate the invasion process.

Similar results were obtained through a DPD simulation (see Supplementary Information File), in which we observed that increased adhesion to the ECM fibres induces an increased basal extrusion capacity.

The impact of cell membrane tension caused by distinct mutations on extrusion behaviour was further explored using the phase field model. Corroborating our in vitro findings, we have demonstrated that basal extrusion potential is inversely correlated with membrane tension of the mutant cell. Lower membrane tensions, as seen in the juxtamembrane and intracellular mutants, result in higher extrusion distances (Fig. 3e). These simulations

revealed an interdependence of cell membrane tension and extrusion.

**Tissue curvature is a key regulator of basal cell extrusion.** A remarkable feature of HDGC is the identification of precursor lesions characterised by pagetoid spread of E-cadherin-negative cells below and along the preserved epithelium of gastric glands[10,14]. Hence, we investigated whether the tissue cylindrical structure favours the extrusion phenotype elicited by E-cadherin dysfunction. For that purpose, we have simulated a cylindrical epithelium using both the phase-field and the vertex models.

For the phase-field model, we have followed a similar strategy to that used in the simulation of the flat tissue (Fig. 3). Starting

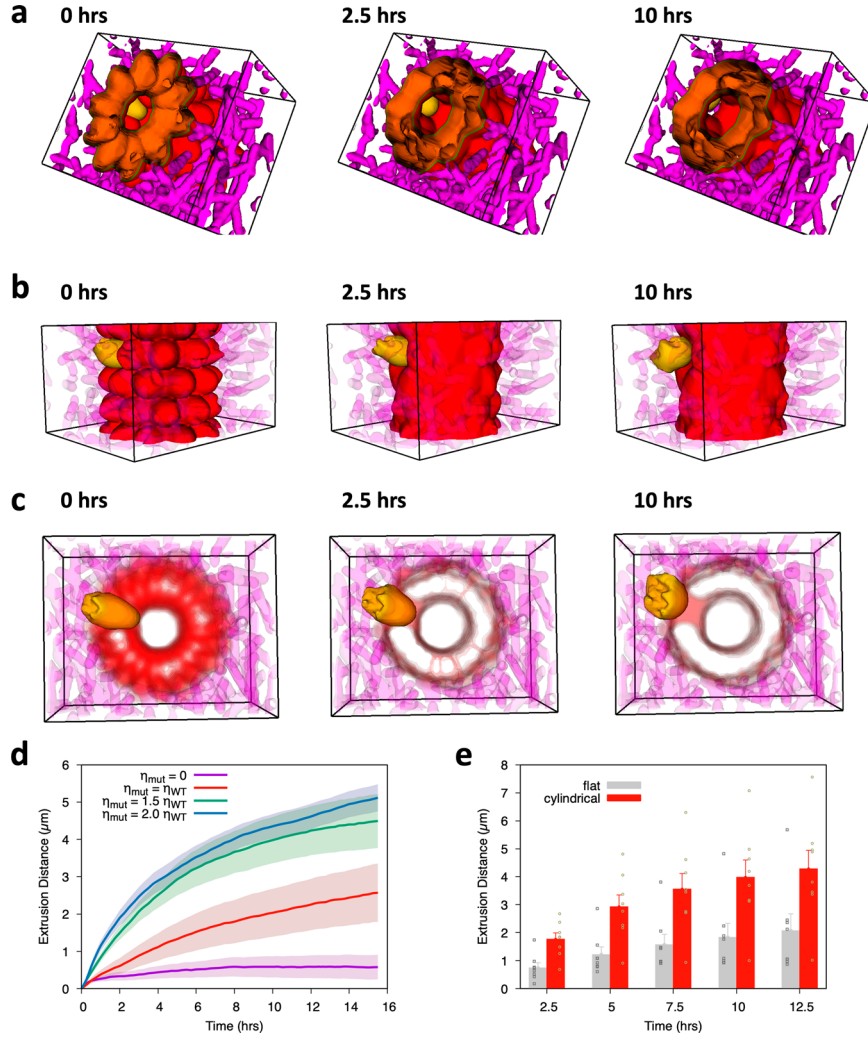

**Fig. 4 Tissue structure impacts the basal cell extrusion. a** Apical view of a typical simulation of an E-cadherin mutant cell in the gland interior, at $t = 0$ h, 2.5 h and 10 h, during the extrusion process. Epithelial cells are marked in red, the mutant cell in yellow, and matrix fibres in magenta. **b** External view of the same simulation at $t = 0$ h, 2.5 h and 10 h. **c** Mutant cell morphology during the extrusion process for the same simulation ($t = 0$ hrs, 2.5 h, and 10 h, respectively). **d** Graph shows the extrusion distance profile of the E-cadherin mutant cell for 16h, when adhesion to ECM is 0, 1, 1.5 and 2 times that of the normal epithelial tissue. For each condition, 8 simulations were performed and standard deviations in the mean extrusion distance are represented by shaded bands. **e** Direct comparison between extrusion distance in cylindrical tissue and the corresponding flat geometry, as a function of time, when the mutant cell adhesion to ECM is equal to 1.5 times that of the wild-type cells. The individual results from the simulations are indicated (in squares for the flat tissue and in circles for the cylindrical geometry), as well as the standard deviation of the mean extrusion distances.

with a configuration of identical elongated cells (with an ideal ellipsoidal shape that soon progresses to a more realistic one) in contact with the matrix, we have considered a single E-cadherin mutant cell unable to establish cell-cell contacts with its normal counterparts (cell-cell adhesion = 0, Fig. 4a–c). Subsequently, we have monitored the fate of the mutant cell when compared to that of the wild-type adhering cells for different mutant cell-ECM adhesion coefficients (Fig. 4d). Similarly to what was observed in the flat tissue, the mutant cell adheres to the ECM fibres to propel itself forward, corroborating the invasion-suppressor function of E-cadherin and the interdependence of cell-cell and cell-ECM linkages. Moreover, our results demonstrated that extrusion distances are higher under a cylindrical architecture, when compared with the flat tissue structure (Figs. 3d and 4d, e). For example, when the adhesion of the mutant cell to the ECM is 1.5 that of the WT cells, at 2.5h and 10h, the mutant cell has travelled 1.8 and 4.0 μm in the cylindrical geometry, whereas in the flat tissue it reached only 0.7 and 2.0 μm, respectively (Fig. 4e). Interestingly, in the cylindrical model, mutant cells are basally

extruded even when cell attachment to the ECM is set to 0, in contrast to what is observed in the flat tissue. This suggests that gland geometry imposes an apical compression to the mutant cell, facilitating cell movement in the basal direction. In addition, as verified in the flat model, cell migration distance increases with its ECM adherence.

To further explore the contribution of tissue curvature to the extrusion phenotype, we have implemented a vertex model in which we varied the perimeter of the cylinder base (Fig. 5a–c). In this system, mutant cells have reduced border tensions, $\Lambda$, representing a weaker intercellular adhesion with their neighbours. For quantitative analysis, the area of the mutant cell was evaluated over time in cylindrical epithelia comprising 10, 15, or 20 cells in perimeter. As shown in Fig. 5d, in a cylinder with a perimeter of 20 cells, the mutant cell area is 29.2 μm² at 2.5 h and 7.4 μm² at 10 h, while in a cylinder with a perimeter of 10 cells, the mutant area is decreased to 6.4 and 2.8 μm², respectively. Accordingly, the tighter the cylinder, i.e., the higher its curvature, the faster the cell area decreases, indicating cell extrusion. We

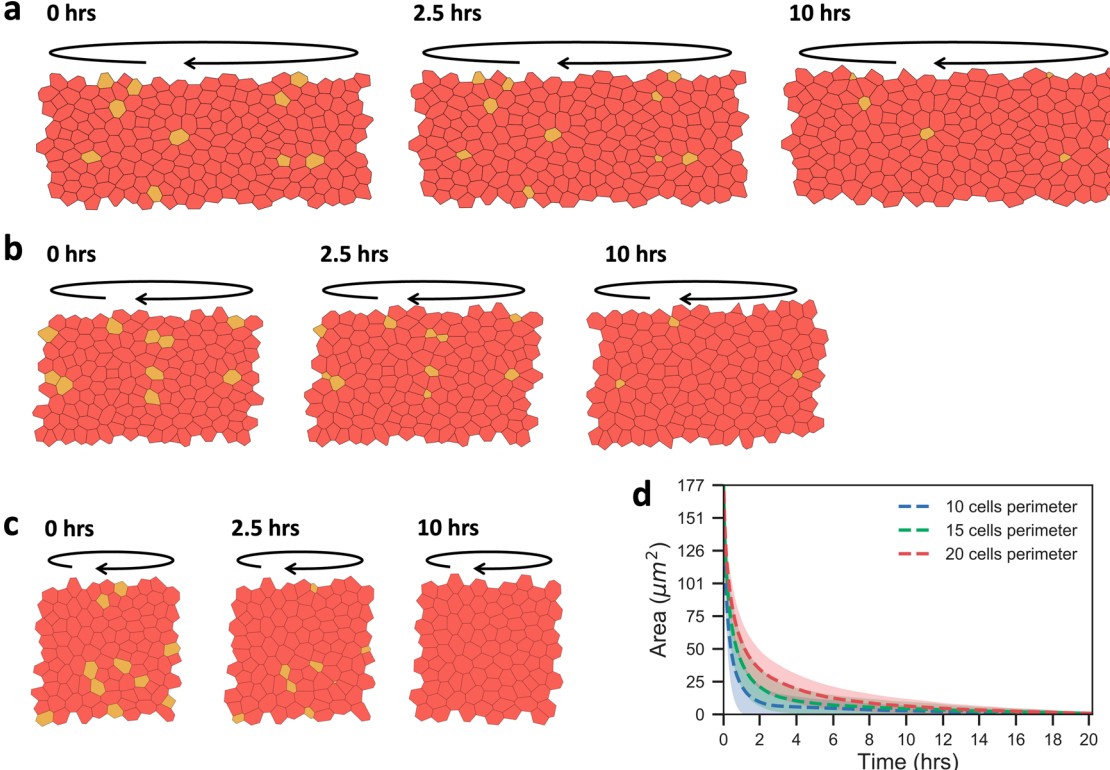

**Fig. 5 Extrusion capacity is dependent on tissue curvature. a** Apical representation of a vertex model simulating a cylindrical epithelium with a base perimeter of 20 cells. Normal epithelial cells are depicted in red whereas mutant cells are represented in yellow. Black circular arrows indicate curvature direction. Left panel illustrates the system state at $t = 0$ h, middle panel shows it at 2.5 h, and right panel at 10 h. **b** Extrusion phenotypes of a cylindrical epithelium with a base perimeter of 15 cells at $t = 0$ h, 2.5 h and 10 h. **c** Results for a cylindrical epithelium with a base perimeter of 10 cells. **d** Graph shows the area of the mutant cells in cylindrical epithelia with bases comprising 10, 15, or 20 cells. Decrease in cell area indicates extrusion. For each condition, 10 simulations were performed and standard deviations in the mean area of mutant cells are represented by shaded bands.

speculate that tissue curvature increases the contact area between the mutant cell and the matrix layer thus promoting its infiltration.

These observations were supported by our DPD simulation (see Supplementary Information File), which confirmed that basal extrusion efficacy is dependent on 3D structure and curvature radius of the tissue.

## Discussion

In this study, we aimed to unravel how loss of E-cadherin activity induces mechanical and biochemical signals, initiating a process of cell extrusion from the normal epithelium to the ECM. Previous work suggests that E-cadherin alterations are associated with increased cell plasticity and with a unique interplay with the ECM, which involves the activation of specific ECM receptors and secretion of attractive ECM components[27,30]. Therefore, we have performed in vitro experiments and developed mathematical simulations to investigate the fate of E-cadherin mutant cells when in close contact with the ECM.

Herein, we have first used a set of cell lines expressing E-cadherin variants identified in HDGC context and affecting different portions of the protein, namely its extracellular, juxtamembrane, and intracellular domains[31–33]. We have demonstrated that E-cadherin mutations compromising the juxtamembrane or the cytoplasmic domain increase cells' ability to basally extrude. We speculate that this is related to weaker coupling with the cytoskeleton, decreasing cell fitness and resistance to the stress imposed by wild-type neighbours[34,35]. Consistent with this, it was recently reported that juxtamembrane and intracellular E-cadherin mutants decrease the capacity of cells to

maintain tensional homoeostasis, while extracellular mutants display higher levels of cytoskeletal tension, with lower variability of contractile moments[36].

Given its influence in cellular behaviour, the ECM has emerged as a key regulator of tumorigenesis[34,37], prompting us to scrutinize the contribution of cell-ECM adhesion in extrusion dynamics. Using both phase-field and DPD simulations, we found that basal extrusion is dependent on a differential cell attachment to ECM. The stronger the linkage with the ECM, the greater the cell capacity to move through the matrix, corroborating previous data demonstrating that ECM provides a ligand site for integrin engagement, generating traction forces and guidance for cancer invasion[38–42]. In contrast, in the absence of cell-ECM interaction, cells leave the epithelium in an apical manner. In support of these results, previous works demonstrate that ECM stiffness promotes integrin clustering and focal adhesion assembly, either disrupting epithelial defence against cancer at the initial pre-malignant stage of carcinogenesis, or promoting cell invasive protrusions at advanced oncogenic stages[42,43]. Our results further confirm that spatial constraints imposed by fibril density and topography also provide physical instruction for cell movement throughout the ECM[44,45].

Importantly, cell extrusion phenotypes, as well as the cell's interplay with the ECM, are highly dependent on tissue context and on microenvironmental factors, which may explain apparent conflicting observations from other studies. For instance, Wee et al. provided evidence that induction of Snail—a well-known transcriptional repressor of E-cadherin—in small cell clusters of breast cancer cells leads to apical extrusion[46]. However, in the cellular model tested, cells expressing a Snail stabilised mutant do

not change canonical EMT markers such as E-cadherin or Vimentin, retaining the ability to mediate cell-cell junctions and assuring epithelial integrity[46]. Consistent with this, Snail stimulation in epithelial monolayers enhanced RhoA signalling and tensile forces at adherens junctions. Transcriptional profiling of Snail cells further revealed a significant downregulation of genes associated with cell-ECM adhesion, namely paxillin, integrins, collagen, and laminin[46]. In contrast, we have previously shown that gastric cancer cells expressing E-cadherin mutations over-express laminin and display abnormal activation of specific integrins, which we believe to promote cell attachment to ECM and an extrusion switch to the basal direction[27,30,47].

Of note, we propose that distinct E-cadherin domains impact differently the extrusion process with alterations in the intracellular portion of E-cadherin enriching basal cell extrusion. Corroborating a domain-associated phenotype is the study by Grieve and Rabouille demonstrating that extracellular cleavage of E-cadherin at the plasma membrane of one epithelial cell drastically affects cell-cell interface and drives apical extrusion[48]. We speculate that disruption of homophilic interactions between E-cadherin molecules on neighbouring cells prevents local cell–cell contact, promoting detachment and apical delamination of the abnormal cell. A new adherens junction is created underneath delaminated cells. The opposite occurs upon disturbing intracellular cadherin-cytoskeletal linkages: there is a decrease in tensional force at the apical side, which is rapidly recovered by apical contraction of wild-type surroundings, forcing cells towards the basal direction.

Motivated by these indications, we have further explored whether variations in cell membrane tension caused by distinct mutations would alter extrusion behaviour. By modulating membrane tension of the mutant cell, we strengthened our in vitro data and attested that lower membrane tensions, as seen in the juxtamembrane and intracellular mutants, synergise with ECM adhesion and result in higher basal extrusion potential.

Aside the role of ECM adhesive features, we revealed that tissue physical cues are determinant for cell's escape from the epithelium. In fact, the cylindrical structure of gastric glands may promote the spread of E-cadherin-negative cells below the normal epithelium. Accordingly, we verified that the extrusion phenotype is correlated with the cylinder perimeter, where a tighter configuration yields a striking extrusive capacity. Tissue curvature supports cell basal extrusion by contracting the apical side of the epithelium while increasing the interface between the mutant cell and the matrix[49,50]. This result explains why neoplastic cells are more frequently found at the neck gland zone, which is more constricted/closed than that of the base gland region[9,10]. Moreover, since glands from different anatomical stomach sections are distinct, we postulate that this underlies predominance of microscopic cancer foci in the proximal stomach, which includes topographical regions, such as cardia, fundus and body[51,52]. Foci of intramucosal carcinoma are less frequently detected in the distal stomach (antrum and pyloric regions)[14,53]. Fundic glands (or oxyntic glands), located in the fundus and body of the stomach, are characterised by straight and narrow tubes, two or more of which open into a single duct. In contrast, pyloric glands are found at the terminal stomach portion and present a shorter and more branched morphology, with a notable lumen. Currently, surveillance protocols for HDGC recommend endoscopic procedures with 28–30 random biopsies, distributed according to the size of the regions (three to five cardia, five fundus, ten body, five transition zone, and five antrum)[9]. However, in the future, geographically targeted biopsies could be considered as an alternative approach.

Mechanical factors that regulate cell extrusion from an epithelial layer and its invasion through the ECM have been previously addressed using computational and mathematical models, although not to the extent of the present work. A 3D vertex model to study the mechanical stability of a monolayer of epithelial cells was developed by Okuda and Fujimoto[54], who concluded that symmetry breaking in the force equilibrium results in cell extrusion. Despite that the model considered adhesive forces between cells, it did not contemplate loss of adhesion as a determinant factor. In our models, by inducing differential adhesion of mutated cells to normal cells and to the ECM, we could replicate biological observations in cancer. Cylindrical, 3D and cross-sectional models were reported by Dunn et al. to describe cell migration in the intestinal crypt[55]. Nonetheless, these models did not take into account the determinant role of cell adhesion to the basement membrane, which in our study was shown to be key for the cell's extrusion ability and velocity. Finally, a cellular Potts model describing the monolayer-to-multilayer transition in epithelial cell sheets, revealed that this shift was enhanced by cell extrusion[56]. The authors discussed the role of cells' mechanical properties and division rate in the extrusion process, but did not explore how this was affected by adhesion. Overall, our work stands out in the mathematical modelling literature by the detailed and extensive computational study of the initial steps of the invasive process mediated by E-cadherin dysfunction. Future studies should address the role of ECM adhesion and tissue curvature in basal extrusion, which remains largely unexplored experimentally.

In conclusion, we propose that, upon loss of E-cadherin, cell intrinsic signals and extrinsic cues cooperate eliciting an effective invasive cancer program. In particular, we have demonstrated that cell-matrix adhesion, low membrane tension, and tissue physical properties, such as cylindrical geometry and surface curvature, promote cell migration towards the basement membrane. This knowledge contributes to the understanding of HDGC aetiology, and highlights the relevance of tissue architecture for the disease phenotypic singularities.

## Methods

**Plasmids**. E-cadherin missense variants p.A634V, p.R749W, and p.V832M were induced in the CDH1pEF6/Myc-His vector, as previously described and established by our group[57]. Each cloning was verified by direct sequencing. The corresponding empty vector and that containing the wild-type cDNA were used as controls in all experiments.

**Cell culture**. AGS cell line (ATCC number: CRL-1739) stably transfected with vectors encoding the wild-type E-cadherin, the variants p.A634V, p.R749W and p.V832M, or the empty vector (Mock) were cultured in RPMI 1640 medium (Gibco, Invitrogen), supplemented with 10% fetal bovine serum (HyClone, Perbio), 1% penicillin/streptomycin (Gibco, Invitrogen) and blasticidin (5 μg/ml; Gibco, Invitrogen). Cells were maintained at 37 °C, under 5% $CO_2$ and humidified air.

**Collagen extrusion assay**. AGS cells stably expressing wild-type E-cadherin and the missense variants p.A634V, p.R749W and p.V832M were resuspended and fluorescently labelled using CellTracker Red CMPTX dye (Invitrogen). Labelled cells were combined with non-labelled wild-type E-cadherin cells at a 1:50 ratio. Cell mixtures were plated at a density of $2.5 \times 10^4$ cell/ml in a μ-Slide Angiogenesis ibiTreat (IBIDI), previously coated with 2 mg/ml neutralised collagen type-I (rat tail, Milipore). Upon achievement of a mature and confluent monolayer, cells were fixed and nuclei were labelled with DAPI (Sigma-Aldrich). Fixation was performed in the dark with 4% formaldehyde in PBS for 20 min, followed by blocking of the aldehyde groups with

50 mM $NH_4Cl$ in PBS for 10 min at room temperature. Subsequently, fixed cells were washed with PBS and permeabilized in 0.2% Triton X-100 for 10 min. Blocking was performed in 1% Bovine Serum Albumin (BSA) for at least 30 min. Cells were incubated with 100 μg/ml of DAPI (Sigma-Aldrich) in the dark for 5 min at room temperature. After washing in PBS, 5–7 μl of IBIDI mounting medium was applied to each condition. Cell-Tracker Red CMPTX and DAPI stained cells were examined using a Leica DMI6000-CS inverted microscope (SP5 II, Leica Microsystems) with HC PL APO CS 40×/1.10 CORR Water objective and Leica Application Suite (LAS) software. Images were collected in 210 nm steps under a z-volume between 15 and 60 μm. For each condition, image triplicates were acquired.

**Image analysis.** To analyse cell infiltration through the collagen matrix, ImageJ software was used to generate a *xz*-section in each acquired image. For quantification purposes, the total number of labelled CellTracker cells was counted and evaluated. Discrimination of basal extrusion, apical extrusion, and epithelial retention was established by CellTracker fluorescence intensity and each cell's nucleus *z* position. Specifically, marked cells with nuclei located above the median epithelial plane were classified as apically extruded, nuclei located along the epithelial plane identified retained cells, and those bellow the reference categorized basally extruded cells. A minimum of 50 marked cells were assessed per condition.

**Statistical analysis.** All data was statistically analysed using the two-tailed unpaired Student's *t*-test from *GraphPad Prism* software (version 8.02). In all analysis, $p \leq 0.05$ was required for statistical significance.

**Phase-field model.** A phase-field model of epithelial tissue was implemented based on models previously described[22,58]. A scalar order parameter was assigned to each individual cell: $\phi_i(\vec{r})$, where $i = 1, 2, \ldots$ identifies the cell. In this way, $\phi_i(\vec{r})$ varies continuously from $\phi_i \approx 1$ inside the cell $i$, to $\phi_i \approx 0$ outside. Likewise, we assigned the continuous order parameter $\phi_0(\vec{r})$ to identify the ECM fibres, such that $\phi_0(\vec{r}) \approx 1$ in the fibres and $\phi_0(\vec{r}) \approx 0$ outside the fibres[22]. Furthermore, we implemented a free energy functional $\mathcal{F}[\phi_0(\vec{r}), \phi_1(\vec{r}), \ldots]$ that describes the cell's membrane tension and the adhesive interaction between cells, and between cells and the ECM fibres[22,58]:

$$
\begin{aligned}
\mathcal{F}[\phi_i(\vec{r})] &= \underbrace{\int \kappa \sum_{i=0}^{N} \left( \frac{1}{4} \phi_i^2 (1 - \phi_i)^2 + \frac{\epsilon^2}{2} (\nabla \phi_i)^2 \right) d\vec{r}}_{F_{\text{interface}}} \\
&+ \underbrace{\int \sum_{<ij>} \left( \gamma h(\phi_i) h(\phi_j) + \eta_{ij} \nabla h(\phi_i) \cdot \nabla h(\phi_j) \right) d\vec{r}}_{F_{\text{interaction}}} \\
&+ \underbrace{\alpha_V \sum_{i=1}^{N} \left( V_{\text{target}} - V[\phi_i] \right)^2}_{F_{\text{volume}}},
\end{aligned}
$$

(1)

where the coefficients $\kappa$ and $\epsilon$ are respectively proportional to the membrane's tension and to the thickness of the interface in the simulation (used as the lattice size in our numerical implementation)[59]. The coefficient $\epsilon$ also plays a role in setting the membrane tension, which in the sharp interface limit is $\sigma = \kappa \frac{\epsilon}{6\sqrt{2}}$. Cell membranes have an intrinsically complex dynamics that depends on their interaction with the cell's cytoskeleton. In contrast to lipid bilayers, which have constant

areas and thus zero or very small surface tensions[60], plasma membranes can extend their area using a range of dynamical processes[61,62], and therefore a membrane tension can be defined. Measurements of cells' membrane tension are within the range 0.003–0.04 mN/m[63–66], and descriptions of cell membrane dynamics driven by membrane tension[67,68] allow to model cell shape while focusing on other effects, such as cell-cell and cell-ECM adhesion.

The coefficient $\gamma$ is proportional to the hard-core energy repulsion between cells (and between cells and ECM fibres), leading to an increase in the system's energy when the cells overlap in space. The sum is carried out over all pairs of distinct order parameters. In this term, the function $h(\phi_i) = \phi_i^2 (3 - 2\phi_i)$ is qualitatively similar to $\phi_i$ in the interval $[0, 1]$, since $h(0) = 0$, $h(1/2) = 1/2$, and $h(1) = 1$, but has extrema at $\phi_i = 1$ and $\phi_i = 0$. Consequently, $h(\phi_i)$ moderates the hard-core repulsion term, which would otherwise be responsible for pushing the order parameter values away from the interval $[0, 1]$[58]. The second term of the interaction part of the system's free energy is proportional to the coefficient $\eta_{ij}$, and describes the surface adhesion between cells and between cells and the ECM fibres. Accordingly, the energy decreases when the two order parameter gradients are anti-parallel, i.e., whenever cells (or cells and ECM) are in contact. In this work, we modulated the coefficients $\eta_{ij}$ to explore the role of cell–cell adhesion in matrix invasion. Ultimately, the volume term penalizes the system energetically when the cell volume is different from its target value $V_{\text{target}}$. The volume of each cell can be directly obtained from the functional $V[\phi_i] = \int h(\phi) d\vec{r}$. The parameter $\alpha_V$ in equation (1) is a penalty parameter, which will be set large enough so that the cell's volume is kept approximately constant during the system's evolution (2).

From the free energy, we obtain the following equation for the time evolution of cells' order parameters, $\phi_i$, $i \geq 1$:

$$
\begin{aligned}
\frac{\partial \phi_i}{\partial t} &= -M \frac{\delta \mathcal{F}}{\delta \phi_i} \\
&= -\frac{1}{\tau} \left[ \frac{1}{2} (1 - \phi_i) \phi_i (1 - 2\phi_i) - \epsilon^2 \nabla^2 \phi_i \right. \\
&\left. + \sum_{j \neq i} \phi_i (1 - \phi_i) \left( \frac{6\gamma}{\kappa} h(\phi_j) - \frac{6\eta_{ij}}{\kappa} \nabla^2 h(\phi_j) \right) \right] \\
&+ 6 M \alpha_V \phi_i (1 - \phi_i) \left( V_{\text{target}} - V[\phi_1] \right)
\end{aligned}
$$

(2)

where $\tau = 1/(M\kappa)$ defines the timescale of the simulation, and $M$ is the system's mobility, chosen to fit the extrusion timescale experimentally observed. In the simulation, the phase-field associated with fibres is static[22]. Model parameters are given in Table 1.

**Vertex dynamics.** A vertex model approach was developed to mimic epithelial tissues, where cell sheets are approximated by

**Table 1 Parameters adopted in the phase-field model.**

| Parameter | Symbol | Value | |
|---|---|---|---|
| Lattice size | $\epsilon$ | 1.0 μm | |
| Average cell diameter | | 15 μm | |
| Cell Volume | $V_{\text{target}}$ | $1.7 \times 10^3$ μm³ | |
| Membrane tension | $\sigma$ | $1.0 \times 10^{-5}$ N/m | [63-66] |
| Interface energy coeff. | $\kappa$ | 85 J/m³ | $\kappa = \frac{6\sigma\sqrt{2}}{\epsilon}$ |
| Hard-core repulsion coeff. | $\gamma$ | 14 J/m³ | Estim. |
| Mobility | $M$ | 5.3 m³J⁻¹s⁻¹ | Fit to exp. |
| Volume conservation coeff. | $\alpha_V$ | 14 J/m⁶ | Estim. |

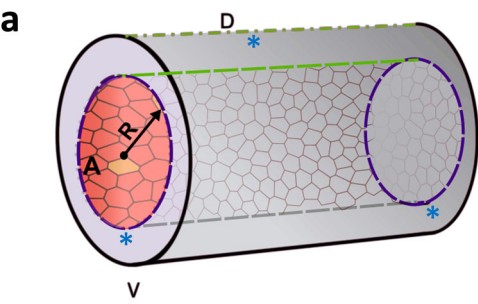

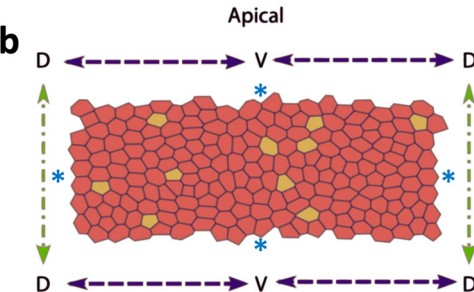

**Fig. 6 Scheme illustrating the Vertex model. a** Cylindrical representation of the tissue structure, where R is the radius and A the apical orientation of the cell sheet, respectively. **b** Apical view of the simulated epithelia. Wild-type cells are represented in red, whereas E-cadherin mutant cells are indicated in yellow. In this open representation of the cylindrical geometry of the tissue, V corresponds to the centre of the simulation domain and D (in green) corresponds to the boundaries. Periodic boundary conditions are implemented in the four boundaries of the domain, indicated by the blue asterisks.

two dimensional tessellations of polygons. In this context, cells can be described as a set of vertices[69–73], and tissue dynamics can be evaluated by formulating the evolution of these vertices as an energy minimisation problem, by the relaxation of the energy function[74]. Of note, this approach permits to consider tailored biophysical energetic terms, accounting for relevant cell properties, namely adhesion and contractibility. We can assume that all vertices follow an over-damped dynamics and cells passively respond to the resulting forces with a viscoelastic behaviour.

To describe the evolution dynamics of the characteristic polygonal morphology of the planar vertex model, forces induced by the prescribed energy potential were encoded into an equation of motion for the $N_v$ vertices describing the tissue, $\mathbf{r} = \{r_i | r_i \in \mathbb{R}^2, i = 1, \dots, N_v\}$, with cells enumerated by $\alpha = 1, \dots, N_c$. A common approach describes the dynamics of each single vertex as a deterministic overdamped equation of motion[75], where inertial terms are neglected, when compared with dissipative terms, and the viscosity parameter, $v$, is constant for each single vertex in the system,

$$\nu \frac{\mathrm{d}r_i}{\mathrm{d}t} = F_i(\mathbf{r}). \tag{3}$$

Here, $F_i(t)$ denotes the total force acting on vertex $i$ at time $t$, and it can be directly derived from the potential energy $E$ as $F_i = -\nabla_i E$. The potential energy has the form[74],

$$E(\mathbf{r_i}) = \sum_\alpha \left[ \frac{K}{2}(A_\alpha - A^0)^2 + \frac{\Gamma}{2} L_\alpha^2 \right] + \sum_{\langle ij \rangle} \Lambda l_{ij}. \tag{4}$$

In this equation, the first sum runs over all the cells, $\alpha$, in the tissue and includes cell-intrinsic energy terms. It considers an elastic term function of the difference between cell area, $A_\alpha$, and the target area, $A^0$, proportional to the elasticity constant $K$ (this term is akin to the $F_{\text{volume}}$ term in the three dimensional phase-field models in equation (1)). A second elastic term depends on

**Table 2 Parameters used in the vertex model.**

| Parameter | Symbol | Value | |
|---|---|---|---|
| Average cell length | | 8.2 μm | |
| Cell target area | $A^0$ | 290 μm² | [69] |
| Cell contractility | $\Gamma/K$ | 0.04 | [74] |
| Line tension | $\Lambda/K$ | $3.7 \times 10^{-9}\,\mathrm{m}^{-1}$ | Estim. |
| Mutant E-cadherin line tension | $\Lambda/K$ | $4.6 \times 10^{-6}\,\mathrm{m}^{-1}$ | Estim. |
| Viscous time-scale | $\nu/K$ | $1.01 \times 10^{9}\,\mathrm{s/m}$ | [74] |

the cell perimeter $L_\alpha$ and describes the contractility of the cell with constant $\Gamma$ (corresponding to the $F_{\text{interface}}$ term in equation (1)). The last sum in (4) runs over all the vertex bonds $\langle ij \rangle$, and implements cell-cell adhesion energy, where $\Lambda$ is a positive constant that represents the line tension. For E-cadherin mutant cells, we reduce the cell-cell adhesion by increasing the line tension, $\Lambda$, in equation (4).

In the present work, we have used the vertex model to simulate a cylindrical gland. Therefore, the domain considered will be a cylinder of radius $R$. Following a previously described implementation[72], equation (3) is written in cylindrical coordinates $(\Theta_i, z_i)$, where $\Theta_i$ is the cylindrical polar angle and $z_i$ is the coordinate along the length of the cylinder for each vertex $i$ (Fig. 6). Moreover, the area of the simulation domain is conserved using an energetic penalisation term[72]. Note that we have imposed periodic boundary conditions in the direction along the length of the cylinder in our simulations. The parameters used for this model are indicated in Table 2.

**Statistics and reproducibility.** The simulation results from Figs. 3, 4 and 5 are the direct result from the integration of equations (2) and (3). For Figs. 3 and 4 the different realisations correspond to different arrangements of the ECM. For Fig. 5 the variability is the result of different initial cell arrangements. Statistics in these plots are given as mean ± standard error of the mean (SEM).

**Reporting summary.** Further information on research design is available in the Nature Portfolio Reporting Summary linked to this article.

**Data availability**
For the experimental data presented in Fig. 2, see Supplementary Data 1.

**Code availability**
The code developed in this study (that was used to obtain the results of Figs. 3, 4 and 5 and the results presented in the Supplementary Information File) is available in https://github.com/phydev/SPiCCAto (Phase Field model), https://bitbucket.org/Pigueco/vertex_model_python_2.7 (Vertex Model), https://github.com/Bordin-Lab/espresso-scripts (DPD model).

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

## Acknowledgements

We dedicate this study to our mentor and loved friend Raquel Seruca. This work was funded by FEDER funds through the Operational Programme Competitiveness Factors—COMPETE and by national funds by FCT—Foundation for Science and Technology under the projects UIDB/04564/2020 and UIDP/04564/2020 (M.M.S., J.C., R.T.), EXPL/MED-ONC/0386/2021 and 2022.02665.PTDC (S.M., F.C., P.C., J.F., R.S.), and by Ministerio de Ciencia, Innovación y Universidades/FEDER (Spain/UE) through grant PID2022-141802NB-I00 (BASIC) (PG). J.F. is funded by the "FCT Scientific Employment Stimulus—Institutional Call" program (CEECINST/00056/2021). We acknowledge the American Association of Patients with Hereditary Gastric Cancer "No Stomach for Cancer" (J.F., R.S.), and the project "P.CCC: Centro Compreensivo de Cancro do Porto" - NORTE-01-0145-FEDER-072678, supported by Norte Portugal Regional Operational Programme (NORTE 2020), under the PORTUGAL 2020 Partnership Agreement, through the European Regional Development Fund (ERDF). We also acknowledge the support of the ALM i3S Scientific Platform, member of the PPBI (PPBI-POCI-01-0145-FEDER-022122). J.R.B. acknowledges the financial support the Brazilian National Council for Scientific and Technological Development (CNPq, proc. 304958/2022-0) and the Research Support Foundation of the State of Rio Grande do Sul (FAPERGS, TO 21/2551-0002024-5). M.M.S. thanks European Union's Horizon 2020 Research and Innovation program under the Marie Skłodowska-Curie Actions Grant, Agreement no. 80113 (Scientia fellowship).

## Author contributions

R.D.M.T., R.S., and J.F. designed and supervised the experiments, modelling and data analyses. F.C., P.C. supervised the experiments, modelling and data analyses. S.M. performed the experiments. P.G., M.M.S., J.R.B., M.B.D., and R.D.M.T. wrote the code, ran the simulations and represented the simulation results. All authors analysed the data, discussed the results, and wrote the manuscript.

## Competing interests

The authors declare no competing interests.
