## [Peer review file · Communications Biology]

The ECM and tissue architecture are major determinants of early invasion mediated by E-cadherin dysfunctionReviewers' comments:

Reviewer #1 (Remarks to the Author):

In this article, Melo and colleagues use a combination of cell culture and different simulations strategies to explain how basal extrusion and invasion occurs in hereditary diffuse gastric cancer. HDGC is associated with localised appearance of basally located cells. HDGC is associated with germline mutation of E-cad, which may be causative for cell extrusion and invasion. The authors first use a cell culture assay with collagen with various mutations of E-cad carried in single cell, which confirmed the increase rate of basal extrusion (specially in mutations affecting strongly the levels of E-cad). They then used a combination of vertex model, phase field model and dissipative particle dynamics model to study the impact of adhesion loss on extrusion. Doing so, they show that a combination of cell-cell adhesion loss coupled to strong ECM adhesion promotes basal extrusion. Interestingly, tissue curvature also promotes basal extrusion, which may explain the very localised impact of HDGC on tubes/gland with small diameters.

The conclusions of the study could have quite broad interest, specially in the field of cell extrusion and cancer biology. So far, the conditions that will either promote basal and apical extrusion are not quite well understood and the contribution of ECM adhesion has been overall quite neglected. Moreover, the authors propose interesting modeling frameworks to study the contribution of different forces to cell extrusion. At this stage though, all the conclusions are based on the results of the simulation and it remains unclear whether this apply experimentally and in vivo. I feel that the conclusion will gain much more interest if could be partially backed up by some experiments, at least with their cell culture assay. I have some suggestions and clarifications that I believe would be worth exploring before publication.

1. If I am correct, the HDGC disease is mostly associated with germline depletion/mutation of E-cad. In that respect, one would not expect mosaic differences in E-cad, but rather a global decrease. As such, how relevant are the simulation and the experiments used by the authors for the disease (since they systematically use one mutant cell surrounded by WT). Could they test in their simulation whether global decrease of cell-cell adhesion can also enhance the rate of basal extrusion ? Do they have any indication that there is heterogeneity of E-cad levels in the patients sample (even from previously published data) ?

2. So far, the conclusion on the link between ECM adhesion and extrusion direction is purely based on simulations. While this is per se a very interesting conclusion, I feel it might be worth exploring this hypothesis at least in the in vitro assay that the authors used. Could they try to impact both E-cad and ECM binding protein in single cells and check whether this promote instead apical extrusion (by depleting core components of ECM binding such as Paxilin or integrins) ? Similarly, could they increase the rate of basal extrusion by strengthening basal adhesion ? Similarly, so far we can only see the rate of basal extrusion in the experiment shown in Figure 3D, what are the rates of apical extrusion in these different contexts ?

3. Several experimental papers have explored the link between E-cad depletion, ECM and cell extrusion. For instance, Grive and Rabouille (JCS 2014 <https://doi.org/10.1242/jcs.147926>) have used a TEV induced depletion of E-cad and showed that this is sufficient to trigger extrusion. Alternatively, Wee et al. (JCS 2020 <https://doi.org/10.1242/jcs.235622>), have shown that Snail induction in single cell (a well known transcriptional repressor of E-cad) promotes basal extrusion through enhanced contractility and reduced ECM adhesion. They actually show that Rho activation and reduction of ECM binding are sufficient to promote basal delamination. It would be important to discuss these results (which partially contradict the conclusions of the theoretical prediction of this article) in the discussion at least.

4. So far, it is a bit difficult to get the intuition of how tissue curvature is actually taken in account in the different model (specially for the vertex model). Could the authors provide a bit more explanation on how this 3D feature is actually encapsulated in the 2D vertex model ?

Other minor points :

- The intro and the abstract never clearly mention that the extrusion are occurring basally (which I suspect is assumed to be equivalent to invasion). I think it would help to clearly mention from the beginning that the authors refer to basal extrusion when mentioning invasion.
- It could be good to clearly show on Figure 1 where is basal and apical
- It is not clear from the methods how the authors counted the delaminated cells experimentally. Is it like in the model ? (based on a threshold distance from the epithelial plane). If so this should be specified in the method.
- In Figure 3D, : is there any apical extrusion in this context ? What is the MOCK condition and why is it higher than WT ?
- End of page 10 : I am not sure it is possible to claim that the capacity to invade relies on E-cad depletion. Basal extrusion is not equal to invasion (which also encapsulate migratory behaviour). Or at least the definition of invasion should be clearly stated by the authors.

Reviewer #2 (Remarks to the Author):

In the manuscript, the authors present study the early mechanical steps of invasions of HDGC cancer syndrome. In particular, the authors focus on the E-cadherin defective cells, which are known to trigger an early invasion mediated by adhesion to the ECM (basal extrusion).

The authors execute both experiment and numerical simulation to unravel the mechanism of the cellular invasion. The manuscript is well written, the figures are self-explanatory and the study deserves publication in my opinion. However, there are some major points that should be clarified. The main one is the fact that the study somehow is presented with no significant crosstalk between the experiments and the simulation. To explain myself, the experiments seem to be used to target which portion(s) of the E-cadherin protein has a major effect on the ability of cells to invade. This experiments reveal interesting results, which are probably relevant to the community. But when the simulations are coupled, there is no mention and link to these mechanistic effects, as the numerical results focus on recapitulating the invasion dynamics, and uncover interesting facts on the geometry/packing of E-cadherin-defective cells and the effect of these geometrical factors on the invasive ability.

In other words, the experiments and the simulations are disconnected. They both deliver interesting results but a common theme is missing. I am not sure this can be fixed by changing the writing, as it seems to me that the two things should be published separately. Maybe an experiment designed to study the curvature effect on invasion ability can provide the missing link.

Moreover there are three different numerical modeling strategies that fundamentally tell the reader the same thing: as far as I understood, in simple words, when cells are defective of to and lose intercellular contacts, they increasingly use the ECM and lead the invasion, and this is amplified by the curvature. The three model all recapitulate this, and it is nice. But my question is: are all these modeling technique needed? Would just the phase field suffice to prove the point? Why do authors chose to simulate the same thing, asking the same question, and getting the same answers from three different models?

Minor details:

"were platted"  were plated.

Fig. 2  it would be good here to indicate where the periodic BCs are.

"mutant cells (labelled in red)"  shown in red?

"the mutated cell is also able to extrude from the epithelial tissue in the basal direction, indicating

that loss of E-cadherin is sufficient to signal basal extrusion."  I don't think there is enough evidence to say so and attribute it only to this mechanical effect, as other pathways could switch and play a role?

"For example, when the adhesion of the mutant cell to the ECM is 1.5 that of the WT cells, at 2.5h and 10h, the mutant cell has travelled 1.8 and 4.0 μm in the cylindrical geometry, whereas in the flat tissue it reached only 0.7 and 2.0 μm , respectively."  This is interesting to me. Again, is it necessary the Vertex model to unravel the mechanism of this?

"The stronger the linkage with ECM, the greater the cell capacity to move through matrix."  There is quite some literature and arguments to cite here, in my opinion, otherwise it just might look obvious to the reader.

Response COMMSBIO-23-0217

We thank the reviewers for their comments and constructive suggestions regarding this manuscript's publication. In this document, we have carefully addressed all points raised by the reviewers. For the sake of clarity, the reviewer's remarks are quoted in blue, followed by our reply.

Reviewer #1:

In this article, Melo and colleagues use a combination of cell culture and different simulations strategies to explain how basal extrusion and invasion occurs in hereditary diffuse gastric cancer. HDGC is associated with localised appearance of basally located cells. HDGC is associated with germline mutation of E-cad, which may be causative for cell extrusion and invasion. The authors first use a cell culture assay with collagen with various mutations of E-cad carried in single cell, which confirmed the increase rate of basal extrusion (specially in mutations affecting strongly the levels of E-cad). They then used a combination of vertex model, phase field model and dissipative particle dynamics model to study the impact of adhesion loss on extrusion. Doing so, they show that a combination of cell-cell adhesion loss coupled to strong ECM adhesion promotes basal extrusion. Interestingly, tissue curvature also promotes basal extrusion, which may explain the very localised impact of HDGC on tubes/gland with small diameters.

The conclusions of the study could have quite broad interest, specially in the field of cell extrusion and cancer biology. So far, the conditions that will either promote basal and apical extrusion are not quite well understood and the contribution of ECM adhesion has been overall quite neglected. Moreover, the authors propose interesting modeling frameworks to study the contribution of different forces to cell extrusion. At this stage though, all the conclusions are based on the results of the simulation and it remains unclear whether this apply experimentally and in vivo. I feel that the conclusion will gain much more interest if could be partially backed up by some experiments, at least with their cell culture assay. I have some suggestions and clarifications that I believe would be worth exploring before publication.

1. If I am correct, the HDGC disease is mostly associated with germline depletion/mutation of E-cad. In that respect, one would not expect mosaic differences in E-cad, but rather a global decrease. As such, how relevant are the simulation and the experiments used by the authors for the disease (since they systematically use one mutant cell surrounded by WT). Could they test in their simulation whether global decrease of cell-cell adhesion can also enhance the rate of basal extrusion? Do they have any indication that there is heterogeneity of E-cad levels in the patients' sample (even from previously published data)?

We acknowledge the reviewer's comments, and we agree that it is important to clarify phenotypic singularities of hereditary diffuse gastric cancer (HDGC), as well as E-cadherin's role in this context. Germline alterations of CDH1 gene, encoding E-cadherin, are a well-known cause of HDGC. As a tumour suppressor gene, inactivation of both alleles of CDH1 is necessary to affect E-cadherin function and cause cancer. Since mutations are inherited from one parent only, every cell from the progeny carries a functional and a defective copy of the gene. Complete gene inactivation occurs upon somatic loss of the wild-type allele, usually by mutation, promoter hypermethylation or chromosomal rearrangements. Depending on the type of CDH1 alteration, E-cadherin expression can vary from complete absence to protein aberrant distribution, which may partially explain differences in disease penetrance and age of onset.

At the early-stages of diffuse gastric cancer, as seen in HDGC, E-cadherin defective cells escape the normal gastric epithelium in an isolated manner and spread diffusely within the mucosa. These single squeezed cells, so called signet ring cells (SRC), are morphologically different and characterized as mucin-filled cells presenting an unusual depolarized nucleus. Occasionally, SRCs are confined within the basement membrane of the glands, below the preserved epithelium of gastric glands, which is widely accepted to be the initial stage of diffuse gastric cancer.

Taking this into account, we engineered a cellular system that mimics random appearance of E-cadherin mutant cells in a normal epithelium, whilst considering the cell's interplay with an extracellular matrix enriched in collagen (a main structural component). We would like to stress out that a continuous epithelium formed by E-cadherin defective cells does not resemble tumour initiation in HDGC.

Following the reviewer's concern, we have now improved the description of the cellular system used and the reason underlying our approach. Accordingly, we have included the following information in page 4:

“In contrast to a continuous epithelium formed by E-cadherin defective cells, this cellular system mimics the random appearance of E-cadherin defective cells in a normal gastric epithelium, according to the

widely accepted model of isolated and diffuse spreading of tumour cells described for the early stages of HDGC [14,29].”

2. So far, the conclusion on the link between ECM adhesion and extrusion direction is purely based on simulations. While this is per se a very interesting conclusion, I feel it might be worth exploring this hypothesis at least in the in vitro assay that the authors used. Could they try to impact both E-cad and ECM binding protein in single cells and check whether this promote instead apical extrusion (by depleting core components of ECM binding such as Paxilin or integrins)? Similarly, could they increase the rate of basal extrusion by strengthening basal adhesion? Similarly, so far we can only see the rate of basal extrusion in the experiment shown in Figure 3D, what are the rates of apical extrusion in these different contexts?

We agree with the reviewer that it would be interesting to evaluate the impact of cell-ECM adhesion through extrusion assays to complement data retrieved from mathematical simulations. For that purpose, we have performed specific inhibition of Integrin $\beta 1$ in R749W E-cadherin mutant cells since we have recently demonstrated that this ECM receptor is abnormally activated in gastric cancer cells with E-cadherin dysfunction, promoting cell scattering and invasive abilities. Unfortunately, we verified that siRNA treatment for Integrin $\beta 1$ was not sufficient to hamper basal extrusion of mutant cells. We speculate that this effect might be related with a potential integrin switch, which enables cells adaptation to unfavourable conditions. Indeed, it has been described that cells increase ECM attachment and force generation by changing their integrin expression profile and activating alternative ECM receptors.

Nevertheless, as suggested by the reviewer, we have included data concerning levels of apical extrusion in Figure 2e. Please see the updated Figure 2 and corresponding caption, as well as the Results section on page 5 of the revised manuscript.

3. Several experimental papers have explored the link between E-cad depletion, ECM and cell extrusion. For instance, Grive and Rabouille (JCS 2014 <https://doi.org/10.1242/jcs.147926>) have used a TEV induced depletion of E-cad and showed that this is sufficient to trigger extrusion. Alternatively, Wee et al. (JCS 2020 <https://doi.org/10.1242/jcs.235622>), have shown that Snail induction in single cell (a well known transcriptional repressor of E-cad) promotes basal extrusion through enhanced contractility and reduced ECM adhesion. They actually show that Rho activation and reduction of ECM binding are sufficient to promote basal delamination. It would be important to discuss these results (which partially contradict the conclusions of the theoretical prediction of this article) in the discussion at least.

In line with the reviewer’s comment, we consider the crosstalk between E-cadherin, ECM attachment, and cell extrusive abilities a critical issue in this manuscript. Therefore, we have improved the discussion of our data in light of the studies referred by the reviewer and other relevant publications addressing this research topic. In fact, cell extrusion phenotypes, as well as the cell’s interplay with the ECM are highly dependent on tissue context and on microenvironmental factors, which may explain different results across studies and apparent conflicting observations. For instance, Wee et al. provided evidence that induction of Snail – a well-known transcriptional repressor of E-cadherin – in small cell clusters of breast cancer cells leads to apical extrusion. However, in the cellular model tested, cells expressing a Snail stabilized mutant do not change

canonical EMT markers such as E-cadherin or Vimentin, and retained the ability to mediate cell-cell junctions, thus assuring epithelial integrity. Consistent with this, Snail stimulation in epithelial monolayers enhanced RhoA signalling and tensile forces at adherens junctions. Transcriptional profiling of Snail cells further revealed a significant downregulation of genes associated with cell-ECM adhesion, namely paxillin, integrins, collagen and laminin. In contrast, we have previously shown that gastric cancer cells expressing E-cadherin mutations overexpress laminin and display abnormal activation of specific integrins, which we believe promote cell attachment to ECM and an extrusion switch to the basal direction.

Importantly, we propose that distinct E-cadherin domains impact differently the extrusion process. Alterations in the intracellular portion of E-cadherin induce basal cell extrusion, whereas defects in the extracellular domain support apical extrusion of mutant cells. Corroborating our data is the study by Grieve & Rabouille demonstrating that extracellular cleavage of E-cadherin at the plasma membrane of one epithelial cell drastically affects cell-cell interface and drives apical extrusion. We speculate that disruption of homophilic interactions between E-cadherin molecules on neighbouring normal cells prevents local cell-cell contact, promoting detachment and apical delamination of the abnormal cell. A new adherens junction is created underneath delaminated cells. A contrasting situation occurs upon disturbing intracellular cadherin-cytoskeletal linkages: there is a decrease in tensional force at the apical side, which is rapidly recovered by apical contraction of wild-type surroundings, forcing cells towards the basal direction.

Please find this information on page 8 of the revised manuscript:

“Importantly, cell extrusion phenotypes, as well as the cell’s interplay with the ECM are highly dependent on tissue context and on microenvironmental factors, which may explain apparent conflicting observations from other studies. For instance, Wee et al. provided evidence that induction of Snail – a well-known transcriptional repressor of E-cadherin – in small cell clusters of breast cancer cells leads to apical extrusion [46]. However, in the cellular model tested, cells expressing a Snail stabilized mutant do not change canonical EMT markers such as E-cadherin or Vimentin, retaining the ability to mediate cell-cell junctions and assuring epithelial integrity [46]. Consistent with this, Snail stimulation in epithelial monolayers enhanced RhoA signalling and tensile forces at adherens junctions. Transcriptional profiling of Snail cells further revealed a significant downregulation of genes associated with cell-ECM adhesion, namely paxillin, integrins, collagen and laminin [46]. In contrast, we have previously shown that gastric cancer cells expressing E-cadherin mutations overexpress laminin and display abnormal activation of specific integrins, which we believe to promote cell attachment to ECM and an extrusion switch to the basal direction [27,30,47].

Of note, we propose that distinct E-cadherin domains impact differently the extrusion process. Alterations in the intracellular portion of E-cadherin induce basal cell extrusion, whereas defects in the extracellular domain support apical extrusion of mutant cells. Corroborating our data is the study by Grieve & Rabouille demonstrating that extracellular cleavage of E-cadherin at the plasma membrane of one epithelial cell drastically affects cell-cell interface and drives apical extrusion [48]. We speculate that disruption of homophilic interactions between E-cadherin molecules on neighbouring cells prevents local cell-cell contact, promoting detachment and apical delamination of the abnormal cell. A new adherens junction is created underneath delaminated cells. The opposite occurs upon disturbing intracellular cadherin-cytoskeletal linkages: there is a decrease in tensional force at the apical side, which is rapidly recovered by apical contraction of wild-type surroundings, forcing cells towards the basal direction.”

After considering these indications, we felt motivated to further explore whether variations in cell membrane tension caused by distinct mutations would alter extrusion behaviour. Therefore, we have used the phase field model to evaluate the interdependence of cell membrane tension and extrusion. Corroborating our in vitro findings, we have demonstrated that basal extrusion potential is inversely correlated with membrane tension of the mutant cell. Lower membrane tensions, as seen in the juxtamembrane and intracellular mutants, result in higher extrusion distances (Fig.3e). These simulations are now included in the Results section, and in the Discussion section.

Results, page 6

The impact of cell membrane tension caused by distinct mutations on extrusion behaviour was further explored using the phase field model. Corroborating our in vitro findings, we have demonstrated that basal extrusion potential is inversely correlated with membrane tension of the mutant cell. Lower membrane tensions, as seen in the juxtamembrane and intracellular mutants, result in higher extrusion distances (Fig. 3e). These simulations revealed an interdependence of cell membrane tension and extrusion.

Discussion, page 17

Motivated by these indications, we have further explored whether variations in cell membrane tension caused by distinct mutations would alter extrusion behaviour. By modulating membrane tension of the mutant cell, we strengthened our *in vitro* data and attested that lower membrane tensions, as seen in the juxtamembrane and intracellular mutants, synergize with ECM adhesion and result in higher basal extrusion potential.

New plate Fig 3e:

4. So far, it is a bit difficult to get the intuition of how tissue curvature is actually taken in account in the different model (specially for the vertex model). Could the authors provide a bit more explanation on how this 3D feature is actually encapsulated in the 2D vertex model?

To elucidate how the 3D cylindrical geometry is implemented in the vertex model, we have improved Figure 6. In the revised figure, it is now represented the apical surface of the epithelial tissue, which we hope to convey the geometry of the simulation to the reader.

Mathematically, tissue curvature is simulated in the 2D vertex model representation by considering the equations in cylindrical coordinates and setting the area of the domain constant with an energy penalization, as previously described [43]. This is now mentioned on page 14:

“Following a previously described implementation [72], the equation (3) is written in cylinder coordinates (θ_i, z_i) , where θ_i is the cylindrical polar angle and z_i is the coordinate along the length of the cylinder for each vertex i (Fig. 2). Moreover, the area of the simulation domain is conserved using an energetic penalization term [72].”

Other minor points:

- The intro and the abstract never clearly mention that the extrusion are occurring basally (which I suspect is assumed to be equivalent to invasion). I think it would help to clearly mention from the beginning that the authors refer to basal extrusion when mentioning invasion.

In an attempt to clarify the readers regarding this issue, we have now specified basal extrusion in the Abstract and Introduction sections. Moreover, we have highlighted differences between cell extrusion and invasion in the Introduction. Cell extrusion has been defined as a mechanism to control cell number within epithelia and prevent the accumulation of excess cells (overcrowding). Unwanted cells undergo programmed cell death that is followed by apical extrusion or delamination, and ultimately apoptosis. Importantly, evidence has emerged demonstrating that cells may escape this process upon oncogenic transformation. Instead of being eliminated towards the lumen, transformed cells hijack this process and extrude in a basal direction into the stroma, subsequently proliferating and invading adjacent tissues. During invasion, cells migrate and overpass the extracellular matrix through morphological adaptation, activation of specific signalling cascades, and proteolytic degradation of matrix components.

We have added the following paragraph to the Introduction (page 3):

“We envision that these precursor lesions are a manifestation of the cell extrusion process. Cell extrusion has been defined as a mechanism to control cell number within epithelia and prevent the accumulation of excess cells (overcrowding) [15]. Unwanted cells undergo programmed cell death that is followed by apical extrusion or delamination, and ultimately apoptosis [15]. Importantly, evidence has emerged demonstrating that cells may escape apical delamination upon oncogenic transformation [16]. Instead of being eliminated towards the lumen, transformed cells hijack this process and extrude in a basal direction into the stroma, subsequently proliferating and invading adjacent tissues [16]. During invasion, cells migrate and overpass the extracellular matrix (ECM) through morphological adaptation, activation of specific signalling cascades, and proteolytic degradation of matrix components.”

- It could be good to clearly show on Figure 1 where is basal and apical

Following the reviewer’s suggestion, we have now depicted apical and basal surfaces in Figure 1. The corresponding figure legend was updated accordingly.

- It is not clear from the methods how the authors counted the delaminated cells experimentally. Is it like in the model? (based on a threshold distance from the epithelial plane). If so this should be specified in the method.

According to the reviewer’s recommendation, we have improved the description of methods applied for analysis of extruded cells. We specified that discrimination of extrusion phenotypes was established by CellTracker fluorescence intensity and each cell’s nucleus z position. In brief, marked cells with nuclei located above the median epithelial plane were classified as apically extruded, nuclei located along the epithelial plane identified retained cells, and those below the reference categorized basally extruded cells. Please refer to the Methods section, on page 11:

“For quantification purposes, the total number of labelled CellTracker cells was counted and evaluated. Discrimination of basal extrusion, apical extrusion, and epithelial retention was established by CellTracker fluorescence intensity and each cell’s nucleus z position. Specifically, marked cells with nuclei located above the median epithelial plane were classified as apically extruded, nuclei located along the epithelial plane identified retained cells, and those below the reference categorized basally extruded cells. A minimum of 50 marked cells were assessed per condition.”

- In Figure 3D, is there any apical extrusion in this context? What is the MOCK condition and why is it higher than WT?

As requested by the reviewer, we have included in Figure 2e apical extrusion rates evaluated for the same conditions. Please see the new version of Figure 2, and the text on results description, and methods on pages 5 and 10, respectively.

Furthermore, we have also clarified in the caption of Figure 2 that the MOCK condition corresponds to the same cell line transfected with the empty vector and is therefore used as a control condition devoid of E-cadherin expression and function. As such, basal extrusion rates are expected to be higher than in the wild-type condition.

Results, page 5

Regarding apical delamination, the extracellular mutant A634V exhibits increased extrusion levels, when compared with the juxtamembrane and intracellular mutants (Fig. 2e).

Fig. 2 legend was updated with the following sentence:

“The MOCK condition corresponds to the same cell line transfected with the empty vector, and is used as a control condition devoid of E-cadherin expression and function.”

- End of page 10: I am not sure it is possible to claim that the capacity to invade relies on E-cad depletion. Basal extrusion is not equal to invasion (which also encapsulates migratory behaviour). Or at least the definition of invasion should be clearly stated by the authors.

We agree with reviewer that this sentence may create confusion. In the revised manuscript, we have re-phrased this sentence to (page 5 of Results):

“This suggests that extrusion patterns depend on E-cadherin function, as well as on the E-cadherin domain affected.”

More so, as above mentioned, we have now incorporated a paragraph explaining the biological concepts of “basal extrusion” and “invasion”. Please see pages 3-4 of the manuscript.

Reviewer #2 (Remarks to the Author):

In the manuscript, the authors present study the early mechanical steps of invasions of HDGC cancer syndrome. In particular, the authors focus on the E-cadherin defective cells, which are known to trigger an early invasion mediated by adhesion to the ECM (basal extrusion).

The authors execute both experiment and numerical simulation to unravel the mechanism of the cellular invasion. The manuscript is well written, the figures are self-explanatory and the study deserves publication in my opinion. However, there are some major points that should be clarified. The main one is the fact that the study somehow is presented with no significant crosstalk between the experiments and the simulation. To explain myself, the experiments seem to be used to target which portion(s) of the E-cadherin protein has a major effect on the ability of cells to invade. These experiments reveal interesting results, which are probably relevant to the community. But when the simulations are coupled, there is no mention and link to these mechanistic effects, as the numerical results focus on recapitulating the invasion dynamics, and uncover interesting facts on the geometry/packing of E-cadherin-defective cells and the effect of these geometrical factors on the invasive ability.

In other words, the experiments and the simulations are disconnected. They both deliver interesting results but a common theme is missing. I am not sure this can be fixed by changing the writing, as it seems to me that the two things should be published separately. Maybe an experiment designed to study the curvature effect on invasion ability can provide the missing link.

We acknowledge the reviewer's comments and we agree that the link between experiments and simulations could be improved. Therefore, to epitomize our in vitro approach encompassing defects in different domains of E-cadherin, we have now defined sequential conditions of decreased membrane tension using the phase-field model. The extracellular mutant A634V, which is able to reach the membrane, presents a membrane tension close to that of wild-type cells, whereas the mutants R749W and V832M exhibit a more drastic effect [36,57]. With this system, we have demonstrated that basal extrusion potential is inversely correlated with membrane tension of the mutant cell. Lower membrane tensions, as seen in the juxtamembrane and intracellular mutants, result in higher extrusion performance (Fig.3e). These simulations are now included in the Results section on page 6, and in the Discussion section on page 8.

Results, page 6

The impact of cell membrane tension caused by distinct mutations on extrusion behaviour was further explored using the phase field model. Corroborating our *in vitro* findings, we have demonstrated that basal extrusion potential is inversely correlated with membrane tension of the mutant cell. Lower membrane tensions, as seen in the juxtamembrane and intracellular mutants, result in higher extrusion distances (Fig. 3e). These simulations revealed an interdependence of cell membrane tension and extrusion.

Discussion, page 8

“Of note, we propose that distinct E-cadherin domains impact differently the extrusion process. Alterations in the intracellular portion of E-cadherin induce basal cell extrusion, whereas defects in the extracellular domain support apical extrusion of mutant cells. Corroborating our data is the study by Grieve & Rabouille demonstrating that extracellular cleavage of E-cadherin at the plasma membrane of one epithelial cell drastically affects cell-cell interface and drives apical extrusion [48]. We speculate that disruption of homophilic interactions between E-cadherin molecules on neighbouring cells prevents local cell-cell contact, promoting detachment and apical delamination of the abnormal cell. A new adherens junction is created underneath delaminated cells. The opposite occurs upon disturbing intracellular cadherin-cytoskeletal linkages: there is a decrease in tensional force at the apical side, which is rapidly recovered by apical contraction of wild-type surroundings, forcing cells towards the basal direction. Motivated by these indications, we have further explored whether variations in cell membrane tension caused by distinct mutations would alter extrusion behaviour. By modulating membrane tension of the mutant cell, we strengthened our *in vitro* data and attested that lower membrane tensions, as seen in the juxtamembrane and intracellular mutants, synergise with ECM adhesion and result in higher basal extrusion potential.

Moreover there are three different numerical modeling strategies that fundamentally tell the reader the same thing: as far as I understood, in simple words, when cells are defective of to and lose intercellular contacts, they increasingly use the ECM and lead the invasion, and this is amplified by the curvature. The three models all recapitulate this, and it is nice. But my question is: are all these modeling technique needed? Would just the phase field suffice to prove the point? Why do authors chose to simulate the same thing, asking the same question, and getting the same answers from three different models?

In this work, we have explored distinct computational models to demonstrate that results concerning cell behaviour do not depend on the computational approach applied. In fact, we verified that the phase-field, the vertex and the dissipative particle dynamics models agree with the role of cylindrical geometry in induction of cell extrusion upon loss of cell-cell adhesion. Despite that the three models epitomize distinct mechanisms to describe cell shape and movement, they all support the role of cell-ECM adhesion and tissue curvature in extrusion. Whereas the vertex model explores springs and is commonly used to model epithelial tissue dynamics in morphogenesis, the phase-field model depicts cell membrane deformation driven by surface tension and adhesion energies, and the dissipative particle dynamics model is a minimal model of cell movement applying molecular evolution. Accordance among these models strengthens our confidence that the conclusions drawn are independent from each models' intricacies.

Minor details:

"were platted"  were plated.

We appreciate the careful revision and have now corrected the typo (page 10).

Fig. 2  it would be good here to indicate where the periodic BCs are.

As suggested by the reviewer, we have updated Figure 6 and indicated periodic boundary conditions.

"mutant cells (labelled in red)"  shown in red?

We have substituted the term "labelled" to "shown" in page 5.

"the mutated cell is also able to extrude from the epithelial tissue in the basal direction, indicating that loss of E-cadherin is sufficient to signal basal extrusion."  I don't think there is enough evidence to say so and attribute it only to this mechanical effect, as other pathways could switch and play a role?

We acknowledge the reviewer's concern and, accordingly, we have now re-phrased the paragraph in page 5 to highlight that loss of E-cadherin may lead to basal extrusion through either its mechanical load or a myriad of signalling pathways and putative interactors:

"However, when subjected to an adhesion strength equal to its neighbouring cell, the mutated cell is also able to extrude from the epithelial tissue in the basal direction, indicating that loss of E-cadherin can induce basal extrusion through either associated downstream signalling or mechanotransduction outputs."

"For example, when the adhesion of the mutant cell to the ECM is 1.5 that of the WT cells, at 2.5h and 10h, the mutant cell has travelled 1.8 and 4.0 μm in the cylindrical geometry, whereas in the flat tissue it reached only 0.7 and 2.0 μm , respectively."  This is interesting to me. Again, is it necessary the Vertex model to unravel the mechanism of this?

As mentioned above, we have used complementary computational models to address the role of cylindrical geometry in inducing cell extrusion, corroborating our predictions (please see above additional information on this topic).

"The stronger the linkage with ECM, the greater the cell capacity to move through matrix."  There is quite some literature and arguments to cite here, in my opinion, otherwise it just might look obvious to the reader.

We agree with the reviewer's suggestion and have now included supportive literature of our results. Indeed, we postulate that cell-ECM interaction promotes cell capacity to move through the matrix, according to ECM's role as a ligand site for integrin engagement, as a platform for traction force generation, and as a guide for cell migration. Please see the following text added to page 7 and corresponding references:

"The stronger the linkage with the ECM, the greater the cell capacity to move through the matrix, corroborating previous data demonstrating that ECM provides a ligand site for integrin engagement, generating traction forces and guidance for cancer invasion [38-42]."

[38] G. Gritsenko, P., Ilina, O., Friedl, P.: Interstitial guidance of cancer invasion. *The Journal of pathology* 226(2), 185–199 (2012)

[39] Aznavoorian, S., Stracke, M.L., Krutzsch, H., Schiffmann, E., Liotta, L.A.: Signal transduction for chemotaxis and haptotaxis by matrix molecules in tumor cells. *The Journal of cell biology* 110(4), 1427–1438 (1990)

[40] Choquet, D., Felsenfeld, D.P., Sheetz, M.P.: Extracellular matrix rigidity causes strengthening of integrin–cytoskeleton linkages. *Cell* 88(1), 39–48 (1997)

[41] Plotnikov, S.V., Pasapera, A.M., Sabass, B., Waterman, C.M.: Force fluctuations within focal adhesions mediate ecm-rigidity sensing to guide directed cell migration. *Cell* 151(7), 1513–1527 (2012)

[42] Levental, K.R., Yu, H., Kass, L., Lakins, J.N., Egeblad, M., Erler, J.T., Fong, S.F., Csiszar, K., Giaccia, A., Weninger, W., et al.: Matrix crosslinking forces tumor progression by enhancing integrin signaling. *Cell* 139(5), 891–906 (2009)"

We believe that our revision has significantly improved the manuscript, which we hope is now suitable for publication.

I look forward to hearing from you, yours sincerely,

The Authors

REVIEWERS' COMMENTS:

Reviewer #1 (Remarks to the Author):

The authors have significantly improved the manuscript by providing clarifications and important information. I would be positive for publication but I just have three last minor point related to the new data on apical extrusion (Figure 2e) and title/conclusions and would suggest some text modifications.

1. At this stage, the results on apical extrusion seems very hard to interpret. The authors argue that their result fit with the one of C. Rabouille lab (more apical extrusion upon extracellular cleavage of E-cad), but if so, one should see more apical extrusion in the A634V:WT extracellular mutant compared to the WT:WT context. Similarly, it is hard to explain why there is less apical extrusion in the MOCK:WT compared to the WT:WT. If anything, the results in Figure 2e suggest that expression of the A634V rescue extrusion back to WT levels (pretty much same rate apically and basally).

Overall, I would remain much more cautious on the interpretation of these results (which right now are hard to interpret) and I would suggest to change the main text and the discussion accordingly.

2. Since the authors could not recapitulate experimentally the role of ECM on basal extrusion, I would remain cautious in the conclusion and clearly state in the discussion that the role of ECM for basal extrusion remained to be tested experimentally.

3. Related to this point, the article title as it stands suggests that the role of ECM and tissue architecture was also tested *in vitro*, which is not the case (these parameters were only tested in the models). I am enclined to suggest to remove "in vitro" from the title (although this would leave the impression that the article is purely theoretical, but the fact is that the vast majority of conclusions and novelties come from the models).

Reviewer #2 (Remarks to the Author):

The authors have answered my comments mostly in a satisfactory way. I still think that some of the different models used (to strengthen their confidence in the results, I understand) and the relative description can be moved to the supplementary files, and leave only the clearer and most used one (the phase field?). The manuscript is publishable in my opinion.

Comments to the reviewer's remarks

We thank the reviewers for their comments and constructive suggestions regarding the manuscript's publication. In this document, we have addressed minor points raised by the reviewers.

For the sake of clarity, the reviewer's remarks are quoted in blue, followed by our reply.

Reviewer #1

The authors have significantly improved the manuscript by providing clarifications and important information. I would be positive for publication but I just have three last minor point related to the new data on apical extrusion (Figure 2e) and title/conclusions and would suggest some text modifications.

1. At this stage, the results on apical extrusion seems very hard to interpret. The authors argue that their result fit with the one of C. Rabouille lab (more apical extrusion upon extracellular cleavage of E-cad), but if so, one should see more apical extrusion in the A634V:WT extracellular mutant compared to the WT:WT context. Similarly, it is hard to explain why there is less apical extrusion in the MOCK:WT compared to the WT:WT. If anything, the results in Figure 2e suggest that expression of the A634V rescue extrusion back to WT levels (pretty much same rate apically and basally).

Overall, I would remain much more cautious on the interpretation of these results (which right now are hard to interpret) and I would suggest to change the main text and the discussion accordingly.

We thank the reviewer for the positive comments and we agree that apical extrusion results should be interpreted with caution. Accordingly, we have modified the Results and Discussion sections on pages 5 and 8 of the revised manuscript.

On the Results section:

“Regarding apical delamination, the opposite effect is seen. Those conditions depicting lower basal extrusion levels are the ones with higher apical extrusion. The extracellular mutant A634V resembles the wild-type condition and exhibits increased extrusion levels, when compared with the juxtamembrane and intracellular mutants (Fig. 2e).”

On the Discussion section:

“Of note, we propose that distinct E-cadherin domains impact differently the extrusion process with alterations in the intracellular portion of E-cadherin enriching basal cell extrusion. Corroborating a domain-associated phenotype is the study by Grieve & Rabouille demonstrating that extracellular cleavage of E-cadherin at the plasma membrane of one epithelial cell drastically affects cell-cell interface and drives apical extrusion [48].”

2. Since the authors could not recapitulate experimentally the role of ECM on basal extrusion, I would remain cautious in the conclusion and clearly state in the discussion that the role of ECM for basal extrusion remained to be tested experimentally.

We acknowledge the reviewer's concern and, accordingly, we have stated in the Discussion section, on page 9, that the role of ECM adhesion in basal extrusion is a critical issue that we wish to address in the near future using innovative experimental strategies.

“Future studies should address the role of ECM adhesion and tissue curvature in basal extrusion, which remains largely unexplored experimentally.”

3. Related to this point, the article title as it stands suggests that the role of ECM and tissue architecture was also tested in vitro, which is not the case (these parameters were only tested in the models). I am inclined to suggest to remove "in vitro" from the title (although this would leave the impression that the article is purely theoretical, but the fact is that the vast majority of conclusions and novelties come from the models).

In line with the reviewer's comment, we have now altered the manuscript title to **“The ECM and tissue architecture are major determinants of early invasion mediated by E-cadherin dysfunction”**.

Reviewer #2 (Remarks to the Author):

The authors have answered my comments mostly in a satisfactory way. I still think that some of the different models used (to strengthen their confidence in the results, I understand) and the relative description can be moved to the supplementary files, and leave only the clearer and most used one (the phase field?). The manuscript is publishable in my opinion.

We thank the reviewer for his remarks. We believe that all the data presented enriches the article and that accordance among these models strengthens our confidence in the conclusions.